DATA RELEASE

# Genome assembly of the milky mangrove *Excoecaria agallocha*

Hong Kong Biodiversity Genomics Consortium*,†

## ABSTRACT

The milky mangrove *Excoecaria agallocha* is a latex-secreting mangrove that are distributed in tropical and subtropical regions. While its poisonous latex is regarded as a potential source of phytochemicals for biomedical applications, the genomic resources of *E. agallocha* remains limited. Here, we present a chromosomal level genome of *E. agallocha*, assembled from the combination of PacBio long-read sequencing and Omni-C data. The resulting assembly size is 1,332.45 Mb and has high contiguity and completeness with a scaffold N50 of 58.9 Mb and a BUSCO score of 98.4%, with 86.08% of sequences anchored to 18 pseudomolecules. 73,740 protein-coding genes were also predicted. The milky mangrove genome provides a useful resource for further understanding the biosynthesis of phytochemical compounds in *E. agallocha*.

**Subjects** Genetics and Genomics, Plant Genetics, Botany

## INTRODUCTION

The milky mangrove *Excoecaria agallocha* (Euphorbiaceae) (Figure 1A), also known as blind-your-eye mangrove due to its toxic properties causing blindness when its milky latex in contact with eyes, can be found in the brackish water in tropical mangrove forests. In documented human history, this plant has traditionally been used to treat pains and stings from marine organisms, ulcers, as well as leprosy [1, 2]; and is also rich in phytoconstituents and potential source of bioactive compounds, such as polyphenols and terpenoids, for biomedical applications [3, 4]. *E. agallocha* is dioecious, and contrary to typical mangrove species, it does not exhibit specialized aerial roots for gas exchange [5, 6]. It has a relatively wide distribution globally, including Australia, Bangladesh, India, and Hong Kong. While it has important ecological values in mangroves, such as being the food sources of jewel bugs, genome of this ecologically important species is lacking.

## CONTEXT

To date, a few molecular and genomic studies have been conducted on *E. agallocha*. These include a transcriptomic study on the flower sex determination of this dioecious species [7] and the assembly of its chloroplast genome [8]. However, the genome of this mangrove species remained missing. Previous studies have reported different karyotypes of *E. agallocha*, including $2n = 108$ [9], $2n = 130$ [10] and $2n = 140$ [11]. Its reported chromosome numbers were remarkably different to other species in the same genus, such as *Excoecaria acerifolia* Didr. $2n = 24$ [12].

Here, *E. agallocha* (NCBI:txid241838) has been selected as one of the species to be sequenced under the Hong Kong Biodiversity Genomics Consortium (a.k.a. EarthBioGenome Project Hong Kong), formed by researchers from 8 publicly funded universities in Hong Kong, in light to provide a useful resource for further understanding of its biology, ecology, evolution, and to set a foundation to carry out any necessary conservation measures.

**Submitted:** 11 January 2024

* Correspondence on behalf of the consortium. E-mail: jeromehui@cuhk.edu.hk

† Collaborative Authors: Entomological experts who validated the dataset and their affiliations appears at the end of the document

Preprint submitted at https://doi.org/10.1101/2024.01.13.575302

Included in the series: *Hong Kong Biodiversity Genomics* (https://doi.org/10.46471/GIGABYTE_SERIES_0006)

PacBio.hifiasm.asm.bp.p_ctg.gfa.fasta.dedupe.purged.fa.blobtools_pacbio.blobDB.json.bestsum.phylum.p8.span.100.blobplot.bam0



**Figure 1.**  Genome assembly QC and contaminant/cobiont detection.

## METHODS

### Sample collection

Leaf tissues of a male individual of *E. agallocha* were collected at a mangrove sandy shore at Wu Kai Sha, New Territories, Hong Kong (22°25′51.6″N, 114°14′17.3″E) in February 2023. The sample was snap-frozen with liquid nitrogen and stored at −80 °C until DNA extraction.

### High molecular weight DNA extraction

High molecular weight (HMW) genomic DNA isolation was started from grinding 1 g of leaf tissues with liquid nitrogen and performed using NucleoBond HMW DNA kit (Macherey Nagel Item No. 740160.20) with prior CTAB treatment. In brief, around 0.8 g of sample was digested in 5 mL CTAB [13] with addition of 1% PVP for 1 h. After RNAse A treatment, 1/3 volume (~1.6 mL) of 3M potassium acetate was added for contaminant precipitation, followed by two washes of chloroform:IAA (24:1). The resulting supernatant (~4.2 mL) was topped up to 6 ml by adding H1 buffer from NucleoBond HMW DNA kit and continued with the manufacturer's protocol. The resulting DNA was eluted with 80 µL elution buffer (PacBio Ref. No. 101-633-500) and was subject to quality check using the NanoDrop™ One/OneC Microvolume UV–Vis Spectrophotometer, Qubit® Fluorometer, and overnight pulse-field gel electrophoresis.

### Pacbio library preparation and sequencing

Prior to library preparation, DNA shearing was performed. Briefly, a dilution of 5 µg HMW DNA in 120 µL elution buffer was transferred to a g-tube (Covaris Part No. 520079) for 6 passes of centrifugation with 1,990 × g for 2 min, followed by DNA purification with SMRTbell® cleanup beads (PacBio Ref. No. 102158-300). 2 µL sheared DNA was used to perform overnight pulse-field gel electrophoresis while the remaining sheared DNA was stored in a 2 mL DNA LoBind® Tube (Eppendorf Cat. No. 022431048) at 4 °C overnight. Subsequently, a SMRTbell library was constructed using the SMRTbell® prep kit 3.0 (PacBio Ref. No. 102-141-700), following the manufacturer's protocol. In brief, the sheared DNA was processed with DNA repair and then each DNA strand was polished at both ends and tailed with an A-overhang, followed by ligation of T-overhand SMRTbell adapters. The SMRTbell library was purified using SMRTbell® cleanup beads and 2 µL of eluted sample was subject to quantity assessment using Qubit® Fluorometer and fragment size examination with overnight pulse-field gel electrophoresis. After that, a nuclease treatment was processed to eliminate non-SMRTbell structures and a final size-selection step with 35% AMPure PB beads was performed to remove short fragments in the library.

The final library preparation for sequencing was performed with The Sequel® II binding kit 3.2 (PacBio Ref. No. 102-194-100). In brief, the SMRT bell structures were annealed and bound with Sequel II® primer 3.2 and Sequel II® DNA polymerase 2.2, respectively. A final cleanup was processed with SMRTbell® cleanup beads, followed by an addition of serial diluted Sequel II® DNA Internal Control Complex. The library was loaded with the diffusion loading mode at an on-plate concentration of 90 pM. The sequencing was performed on the Pacific Biosciences SEQUEL IIe System running for 30-hour movies with 120 min pre-extension to generate HiFi reads. In total, two SMRT cells were used for the sequencing. Details of the resulting sequencing data are listed in Table 1.

**Table 1.** Summary of genomic sequencing data.

| Library | Reads | Bases | Coverage (X) | Accession number |
|---|---|---|---|---|
| PacBio HiFi | 3,503,202 | 33,204,508,502 | 25 | SRR24631716 |
| Omnic | 500,462,964 | 75,069,444,600 | 56 | SRR26908863 |

## Omnic-C library preparation and sequencing

A nuclei isolation procedure was performed from 2 g ground leaf tissues, following the modification of Workman *et al.* [14]. The resulting nuclei pellet was used to construct an Omni-C library using the Dovetail® Omni-C® Library Preparation Kit (Dovetail Cat. No. 21005) by following the manufacturer's instructions. In brief, the nuclei pellet was resuspended in 4 mL 1× PBS, followed by crosslinking with formaldehyde and DNA digestion with endonuclease DNase I. The concentration and fragment size of the digested lysate was quantified using Qubit® Fluorometer and TapeStation D5000 HS ScreenTape, respectively. Subsequently, both ends of DNA were polished and ligation of biotinylated bridge adaptors were proceeded at 22 °C for 30 min. Proximity ligation between crosslinked DNA fragments was conducted at 22 °C for 1 h, followed by crosslink reversal of DNA and then purification with SPRIselect™ Beads (Beckman Coulter Product No. B23317).

End repair and adapter ligation were conducted using the Dovetail™ Library Module for Illumina (Dovetail Cat. No. 21004). In brief, DNA was tailed with an A-overhang and then ligated with Illumina-compatible adapters at 20 °C for 15 min. The Omni-C library was sheared using USER Enzyme Mix and purified with SPRIselect™ Beads. Afterwards, the DNA fragments were isolated using Streptavidin Beads. The DNA library was amplified with Universal and Index PCR Primers from the Dovetail™ Primer Set for Illumina (Dovetail Cat. No. 25005). A final size selection step was done with SPRIselect™ Beads to retain DNA fragments ranging between 350 bp and 1000 bp only. The concentration and fragment size of the library was validated by Qubit® Fluorometer and TapeStation D5000 HS ScreenTape, respectively. The qualified library was eventually sequenced on an Illumina HiSeq-PE150 platform. Details of the resulting sequencing data are listed in Table 1.

## Genome assembly and gene model prediction

*De novo* genome assembly was performed using Hifiasm (version 0.16.1-r375) [15] with default parameters, which was then searched against the NT database using BLAST for the input for BlobTools (v1.1.1) [16] with default parameters to identify and remove any possible contaminations (Figure 1). Haplotypic duplications were removed using "purge_dups" based on the depth of HiFi reads [17] with default parameters. Furthermore, proximity ligation data sequenced from the Omni-C library were employed to scaffold the assembly with YaHS (version 1.2a.2) [18] with default parameters.

Gene model prediction was run by funannotate (version 1.8.15) [19] using the following parameters "--repeats2evm --protein_evidence uniprot_sprot.fasta --genemark_mode ES --optimize_augustus --organism other --max_intronlen 350000". The predicted gene models from various prediction sources including GeneMark, high-quality Augustus predictions (HiQ), pasa, Augustus, GlimmerHM and snap were combined and processed with Evidence Modeler to produce the annotation files.

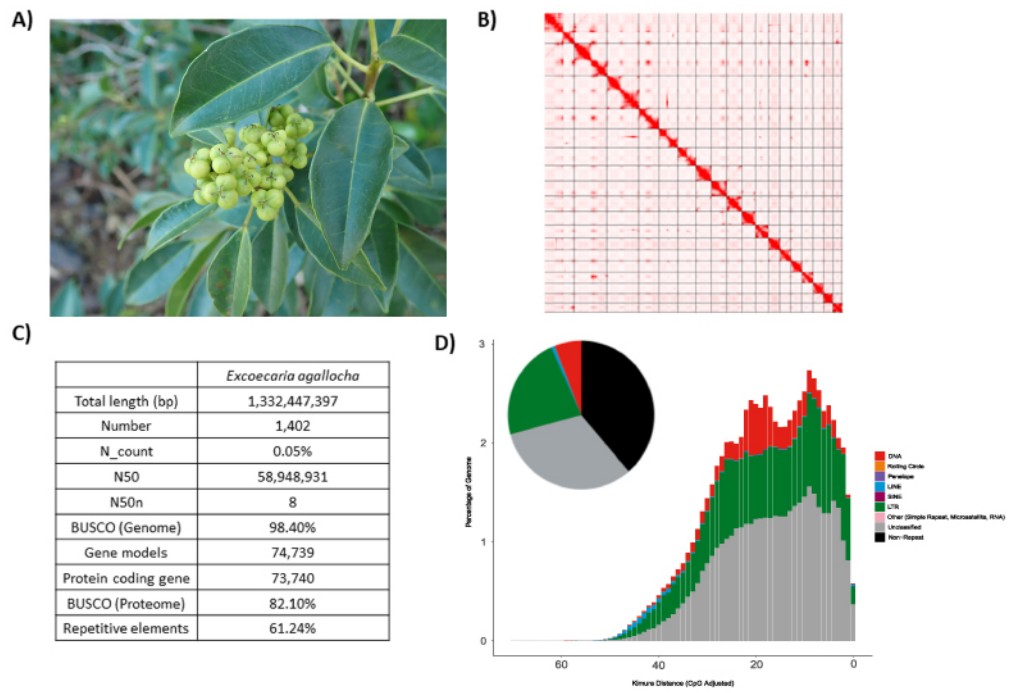

**Figure 2.** Genomic information of *Excoecaria agallocha*. (A) Picture of a female *Excoecaria agallocha*; (B) Genome statistics; (C) Omni-C contact map of the assembly; (D) Pie chart and repeat landscape plot of repetitive elements in the assembled genome annotated by Earl Grey.

## Repeat annotation

Annotation of transposable elements (TEs) were conducted using the Earl Grey TE annotation workflow pipeline (version 1.2) [20] as well as the Extensive de novo TE Annotator (EDTA) (v2.2.0) [21].

## RESULTS AND DISCUSSION

## Genome assembly of Excoecaria agallocha

A total of 33.20 Gb of HiFi reads from the whole genome of milky mangrove *Excoecaria agallocha* were generated by PacBio sequencing. After scaffolding with 75 Gb Omni-C data, 86.08% of the sequences were assembled into 18 pseudochromosomes (Figure 2B). The assembled genome size was 1,332.45 Mb, with 1,402 scaffolds and a scaffold N50 of 58.95 Mb. The complete BUSCO value was estimated to be 98.4% (viridiplantae_odb10) (Figure 2C; Table 2). The GC content was 32.17%. A total of 73,740 protein-coding genes were predicted, with a mean coding sequence length of 288 amino acids (AA) and a BUSCO score of 82.1% (Figure 2C).

Repeat content analyses revealed that transposable elements (TEs) account for approximately 40%–60% of the milky mangrove assembly by annotation tools EDTA and Earl Grey, respectively. The results and classifications of TEs are summarized in Figure 2D and Table 3.

**Table 2.** Genome statistic and sequencing information.

| | *Excoecaria agallocha* |
|---|---|
| Total length (bp) | 1,332,447,397 |
| Number | 1,402 |
| Mean length (bp) | 950,390 |
| Longest | 126,146,454 |
| Shortest | 1,000 |
| N_count | 668,400 |
| Gaps | 3,342 |
| N50 | 58,948,931 |
| N50n | 8 |
| N70 | 43,065,745 |
| N70n | 13 |
| N90 | 1,110,898 |
| N90n | 41 |
| BUSCO (Genome) | C:98.4% [S:39.3%, D:59.1%], F:0.5%, M:1.1%, n:425 |
| BUSCO (Proteome) | C:82.1% [S:45.9%, D:36.2%], F:12.0%, M:5.9%, n:425 |
| HiFi Reads | 3,503,202 |
| HiFi Bases | 33,204,508,502 |
| HiFi Q30% | 3 |
| HiFi Q20% | 4 |
| HiFi GC% | 32 |
| HiFi Nppm | 0 |
| HiFi Ave_len | 9,478 |
| HiFi Min_len | 153 |
| HiFi Max_len | 33,856 |

**Table 3.** Summary of transposable element annotation by Earl Grey and EDTA.

**Repeat analsysis by EarlGrey**

| Classification | Total length (bp) | Count | Proportion (%) | No. of distinct classifications |
|---|---|---|---|---|
| DNA | 78,065,183 | 77,115 | 5.86 | 7,410 |
| LINE | 10,652,399 | 10,771 | 0.80 | 3,831 |
| LTR | 299,391,089 | 124,737 | 22.47 | 7,829 |
| Other (Simple Repeat, Microsatellite, RNA) | 1,141,160 | 932 | 0.09 | 349 |
| Penelope | 37,879 | 151 | 0.00 | 137 |
| Rolling Circle | 1,147,084 | 1,803 | 0.09 | 1,028 |
| SINE | 25,619 | 107 | 0.00 | 93 |
| Unclassified | 425,523,922 | 338,107 | 31.94 | 8,243 |
| SUM | 815,984,335 | 553,723 | 61.24 | 28,920 |

**Repeat analysis by EDTA**

| Classification | | bp masked | Count | % masked |
|---|---|---|---|---|
| LINE | L1 | 15,082,662 | 47,152 | 1.13 |
| | L2 | 328,592 | 714 | 0.02 |
| | RTE | 398,335 | 1,397 | 0.03 |
| LTR | Copia | 65,202,479 | 48,709 | 4.90 |
| | Gypsy | 41,881,198 | 47,938 | 3.14 |
| | Unknown | 2,526 | 20 | 0.00 |
| TIR | CACTA | 48,274,154 | 150,680 | 3.62 |
| | Mutator | 96,860,270 | 244,712 | 7.27 |
| | PIF Harbinger | 51,260,554 | 123,517 | 3.85 |
| | Tc1 Mariner | 39,900,854 | 48,297 | 3.00 |
| | hAT | 29,307,586 | 70,668 | 2.20 |
| Non-LTR | Penelope | 10,803,234 | 38,103 | 0.81 |
| Non-TIR | helitron | 74,727,754 | 205,818 | 5.61 |
| Repeat region | | 57,365,013 | 206,847 | 4.31 |
| Total interspersed | | 531,395,211 | 1,234,572 | 39.90 |

**Table 4.** Information of 18 pseudochromosomes and BUSCO result.

| Chr Number | scaffold_length | scaffold_id | % of whole genome |
|---|---|---|---|
| 1 | 126,146,454 | scaffold_2_1 | 9.47% |
| 2 | 125,683,813 | scaffold_3_1 | 9.43% |
| 3 | 115,235,106 | scaffold_1_1 | 8.65% |
| 4 | 75,925,046 | scaffold_4_1 | 5.70% |
| 5 | 71,251,965 | scaffold_5_1 | 5.35% |
| 6 | 70,346,551 | scaffold_6_1 | 5.28% |
| 7 | 59,451,589 | scaffold_7_1 | 4.46% |
| 8 | 58,948,931 | scaffold_8_1 | 4.42% |
| 9 | 57,619,229 | scaffold_9_1 | 4.32% |
| 10 | 52,150,882 | scaffold_10_1 | 3.91% |
| 11 | 48,057,327 | scaffold_11_1 | 3.61% |
| 12 | 46,431,560 | scaffold_12_1 | 3.48% |
| 13 | 43,065,745 | scaffold_13_1 | 3.23% |
| 14 | 41,974,720 | scaffold_14_1 | 3.15% |
| 15 | 41,514,226 | scaffold_15_1 | 3.12% |
| 16 | 40,341,847 | scaffold_16_1 | 3.03% |
| 17 | 37,296,423 | scaffold_17_1 | 2.80% |
| 18 | 35,487,587 | scaffold_18_1 | 2.66% |
| SUM | 1,146,929,001 | | 86.08% |
| BUSCO | C:92.7% [S:35.3%, D:57.1%] | | |

## CONCLUSION AND FUTURE PERSPECTIVE

The genome assembly of *E. agallocha* presented in this study is the first genomic resource for this mangrove species, which provides a valuable resource for further investigation in the biosynthesis of phytochemical compounds in its milky latex and for the understanding of biology and evolution in genome architecture in the Euphorbiaceae family.

## DATA VALIDATION AND QUALITY CONTROL

During HMW DNA extraction and Pacbio library preparation, quality control of the sample or library was assessed with the NanoDrop™ One/OneC Microvolume UV–Vis Spectrophotometer, Qubit® Fluorometer, and overnight pulse-field gel electrophoresis. The Omni-C library was validated by Qubit® Fluorometer and TapeStation D5000 HS ScreenTape.

For the genome assembly, BlobTools (v1.1.1) [16] was used to identify and remove any possible contaminations (Figure 1). Benchmarking Universal Single-Copy Orthologs (BUSCO, v5.5.0) [22] was run with a collection of single-copy orthologs dataset for the Viridiplantae (Viridiplantae Odb10) to validate the completeness of the genome assembly and gene annotation (Table 2; Table 4).

## DISCLAIMER

The genomic data generated in this study was not assessed for the potential level of polyploidy.

## DATA AVAILABILITY

The raw reads generated in this study were deposited in the NCBI database under the SRA accession SRR24631716 and SRR26908863. The genome, genome annotation and repeat annotation files were made publicly available in Figshare [23].

## ABBREVIATIONS

BUSCO: Benchmarking Universal Single-Copy Orthologs; HMW: High Molecular Weight; TE: Transposable Element.

## DECLARATIONS

### Ethics approval and consent to participate

The authors declare that ethical approval was not required for this type of research.

### Competing interests

The authors declare that they do not have competing interests.

### Authors' contribution

JHLH, TFC, LLC, SGC, CCC, JKHF, JDG, SCKL, YHS, CKCW, KYLY and YW conceived and supervised the study; STSL, DTWL and JSYL collected the samples; STSL and WLS carried out DNA extraction, library preparation and genome sequencing; HYY arranged the logistics of samples; WN performed genome assembly and gene model prediction.

### Funding

This work was funded and supported by the Hong Kong Research Grant Council Collaborative Research Fund (C4015-20EF), CUHK Strategic Seed Funding for Collaborative Research Scheme (3133356) and CUHK Group Research Scheme (3110154).

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

## DETAILS OF COLLABORATIVE AUTHORS

### • List of authors in Hong Kong Biodiversity Genomics Consortium

Jerome H. L. Hui,[1] Ting Fung Chan,[2] Leo Lai Chan,[3] Siu Gin Cheung,[4] Chi Chiu Cheang,[5,6] James Kar-Hei Fang,[7] Juan Diego Gaitan-Espitia,[8] Stanley Chun Kwan Lau,[9] Yik Hei Sung,[10,11] Chris Kong Chu Wong,[12] Kevin Yuk-Lap Yip,[13,14] Yingying Wei,[15] Sean Tsz Sum Law,[1] Wai Lok So,[1] Wenyan Nong,[1] David Tai Wai Lau,[16] Shing Yip Lee,[17] Ho Yin Yip[1]

[1]School of Life Sciences, Simon F.S. Li Marine Science Laboratory, State Key Laboratory of Agrobiotechnology, Institute of Environment, Energy and Sustainability, The Chinese University of Hong Kong, Hong Kong, China

[2]School of Life Sciences, State Key Laboratory of Agrobiotechnology, The Chinese University of Hong Kong, Hong Kong SAR, China

[3]State Key Laboratory of Marine Pollution and Department of Biomedical Sciences, City University of Hong Kong, Hong Kong SAR, China

[4]State Key Laboratory of Marine Pollution and Department of Chemistry, City University of Hong Kong, Hong Kong SAR, China

[5]Department of Science and Environmental Studies, The Education University of Hong Kong, Hong Kong SAR, China

[6]EcoEdu PEI, Charlottetown, PE, C1A 4B7, Canada

[7]Department of Food Science and Nutrition, Research Institute for Future Food, and State Key Laboratory of Marine Pollution, The Hong Kong Polytechnic University, Hong Kong SAR, China

[8]The Swire Institute of Marine Science and School of Biological Sciences, The University of Hong Kong, Hong Kong SAR, China

[9]Department of Ocean Science, The Hong Kong University of Science and Technology, Hong Kong SAR, China

[10]Science Unit, Lingnan University, Hong Kong SAR, China

[11]School of Allied Health Sciences, University of Suffolk, Ipswich, IP4 1QJ, UK

[12]Croucher Institute for Environmental Sciences, and Department of Biology, Hong Kong Baptist University, Hong Kong SAR, China

[13]Department of Computer Science and Engineering, The Chinese University of Hong Kong, Hong Kong SAR, China

[14]Sanford Burnham Prebys Medical Discovery Institute, La Jolla, CA, USA

[15]Department of Statistics, The Chinese University of Hong Kong, Hong Kong SAR, China

[16]Shiu-Ying Hu Herbarium, School of Life Sciences, The Chinese University of Hong Kong, Hong Kong SAR, China

[17]Simon F.S. Li Marine Science Laboratory, The Chinese University of Hong Kong, Hong Kong SAR, China

