## [Reviewer Report]

Comments on revised manuscriptThe author has responded well to the previous comments and the manuscript is now suitable for publication. Regarding the issue of gene annotation, the completeness of annotation in version 2 has significantly improved. However, the author is concerned about the large number of genes. Therefore, here I propose a solution for filtering gene models: you could filter out incomplete gene models and gene models overlapping with repeats if the overlap ratio of CDS region were more than 80%. For genes with CDS lengths less than 150 bp or less than 750 bp and 3 CDS, you could used Hmmer with the Pfam database for validation. If no alignment result was obtained or the alignment coverage was less than 25%, the gene model could be filtered out.

---

## [Editor Report]

Editor’s AssessmentThis work is part of a series of papers from the Hong Kong Biodiversity Genomics Consortium sequencing the rich biodiversity of species in Hong Kong. This example assembles the genome of the milky mangrove Excoecaria agallocha, also known as blind-your-eye mangrove due to its toxic properties of its milky latex that can cause blindness when it comes into contact with the eyes. Living in the brackish water of tropical mangrove forests from India to Australia, they are an extremely important habitat for a diverse variety of aquatic species, including the mangrove jewel bug of which this tree is the sole food source for the larvae. Using PacBio HiFi long-reads and Omni-C technology a 1,332.45 Mb genome was assembled, with 1,402 scaffolds and a scaffold N50 of 58.95 Mb. After feedback the annotations were improved, predicting a very high number (73,740) protein coding genes. The data presented here provides a valuable resource for further investigation in the biosynthesis of phytochemical compounds in its milky latex with the potential of many medicinal and pharmacological properties. As well as increasing the understanding of biology and evolution in genome architecture in the Euphorbiaceae family and mangrove species adapted to high levels of salinity.

---

## [Reviewer Report]

Reviewer name and names of any other individual's who aided in reviewer Minghui KangDo you understand and agree to our policy of having open and named reviews, and having your review included with the published papers. (If no, please inform the editor that you cannot review this manuscript.)YesIs the language of sufficient quality?YesPlease add additional comments on language quality to clarify if needed
Are all data available and do they match the descriptions in the paper? YesAdditional CommentsAre the data and metadata consistent with relevant minimum information or reporting standards? See GigaDB checklists for examples <a href="http://gigadb.org/site/guide" target="_blank">http://gigadb.org/site/guide</a>YesAdditional CommentsIs the data acquisition clear, complete and methodologically sound?YesAdditional CommentsThe sample collection site needs to include latitude and longitude data.Is there sufficient detail in the methods and data-processing steps to allow reproduction?NoAdditional CommentsPlease add the software version number to all the software mentioned in the manuscript. Additionally, if the software uses default parameters, please provide the corresponding description. If specific parameters are used, please indicate the corresponding parameters.Is there sufficient data validation and statistical analyses of data quality? YesAdditional CommentsIs the validation suitable for this type of data?YesAdditional CommentsIs there sufficient information for others to reuse this dataset or integrate it with other data?YesAdditional CommentsAny Additional Overall Comments to the AuthorThis study presents the assembly of an Excoecaria agallocha genome using PacBio HiFi and Omni-C technologies. The assembly exhibits good contiguity and completeness, providing a valuable resource for further understanding the phylogenetic position, evolutionary history, and natural product biosynthesis in Excoecaria agallocha. However, there are still some issues that need to be addressed and modified, including the following points:  L82 It would be preferable to mention the number of chromosomes and the anchor rate of the chromosome-scale assembly here, as well as the estimated genome size based on K-mer analysis, to further support the accuracy and completeness of the assembly.  L88 I think the authors need to rearrange the order of the figures, as it is not appropriate for Fig. 1F to appear before Fig. 1A. Please check the results part and arrange the pictures in a reasonable order.  L117 The sample collection site needs to include latitude and longitude data.  L187 Please add the software version number to all the software mentioned in the manuscript. Additionally, if the software uses default parameters, please provide the corresponding description. If specific parameters are used, please indicate the corresponding parameters.  L219 The pseudochromosome scaffolding rate of 86.08% appears to be somewhat low (<90%). The sequences that were not scaffolded onto chromosomes could be a result of untrimmed redundancy in the genome assembly or could indicate some assembly errors.  L220 Please note that in this instance, Fig. 1C appears before Fig. 1B in the text. I kindly request the author to review and adjust the numbering and arrangement of figures throughout the entire manuscript.  L223 The quality of gene annotation appears to be significantly lower than the quality of genome assembly (82.1%/98.4%), indicating poor gene annotation accuracy. Please review the accuracy of the HMM model trained by the Augustus software or consider using a more accurate annotation workflow.  L225 Unclassified repetitive sequences account for over 50% of the total repetitive sequences, which can significantly impact subsequent analyses relying on repetitive sequences. It is recommended to use alternative software, such as The Extensive de novo TE Annotator (EDTA), which provides more accurate classification and utilizes a more comprehensive repetitive sequence library, to validate these results.RecommendationMajor Revision

---

## [Reviewer Report]

Reviewer name and names of any other individual's who aided in reviewer Jarkko SalojarviDo you understand and agree to our policy of having open and named reviews, and having your review included with the published papers. (If no, please inform the editor that you cannot review this manuscript.)YesIs the language of sufficient quality?YesPlease add additional comments on language quality to clarify if needed
Are all data available and do they match the descriptions in the paper? YesAdditional CommentsAre the data and metadata consistent with relevant minimum information or reporting standards? See GigaDB checklists for examples <a href="http://gigadb.org/site/guide" target="_blank">http://gigadb.org/site/guide</a>YesAdditional CommentsIs the data acquisition clear, complete and methodologically sound?YesAdditional CommentsIs there sufficient detail in the methods and data-processing steps to allow reproduction?YesAdditional CommentsIs there sufficient data validation and statistical analyses of data quality? YesAdditional CommentsIs the validation suitable for this type of data?YesAdditional CommentsIs there sufficient information for others to reuse this dataset or integrate it with other data?YesAdditional CommentsAny Additional Overall Comments to the AuthorRecommendationAccept